# Purinergic–Glycinergic Interaction in Neurodegenerative and Neuroinflammatory Disorders of the Retina

**DOI:** 10.3390/ijms22126209

**Published:** 2021-06-08

**Authors:** Laszlo G. Harsing, Gábor Szénási, Tibor Zelles, László Köles

**Affiliations:** 1Department of Pharmacology and Pharmacotherapy, Semmelweis University, H-1089 Budapest, Hungary; zelles.tibor@med.semmelweis-univ.hu (T.Z.); koles.laszlo@med.semmelweis-univ.hu (L.K.); 2Institute of Translational Medicine, Semmelweis University, H-1089 Budapest, Hungary; szenasi.gabor@med.semmelweis-univ.hu; 3Department of Oral Biology, Semmelweis University, H-1089 Budapest, Hungary

**Keywords:** retina, purinergic modulation, glycinergic neurotransmission, microglia, neuroinflammation, neurodegeneration, glycine transporters

## Abstract

Neurodegenerative–neuroinflammatory disorders of the retina seriously hamper human vision. In searching for key factors that contribute to the development of these pathologies, we considered potential interactions among purinergic neuromodulation, glycinergic neurotransmission, and microglia activity in the retina. Energy deprivation at cellular levels is mainly due to impaired blood circulation leading to increased release of ATP and adenosine as well as glutamate and glycine. Interactions between these modulators and neurotransmitters are manifold. First, P2Y purinoceptor agonists facilitate reuptake of glycine by glycine transporter 1, while its inhibitors reduce reverse-mode operation; these events may lower extracellular glycine levels. The consequential changes in extracellular glycine concentration can lead to parallel changes in the activity of NR1/NR2B type NMDA receptors of which glycine is a mandatory agonist, and thereby may reduce neurodegenerative events in the retina. Second, P2Y purinoceptor agonists and glycine transporter 1 inhibitors may indirectly inhibit microglia activity by decreasing neuronal or glial glycine release in energy-compromised retina. These inhibitions may have a role in microglia activation, which is present during development and progression of neurodegenerative disorders such as glaucomatous and diabetic retinopathies and age-related macular degeneration or loss of retinal neurons caused by thromboembolic events. We have hypothesized that glycine transporter 1 inhibitors and P2Y purinoceptor agonists may have therapeutic importance in neurodegenerative–neuroinflammatory disorders of the retina by decreasing NR1/NR2B NMDA receptor activity and production and release of a series of proinflammatory cytokines from microglial cells.

## 1. Introduction

Neurodegenerative disorders occur with a high incidence in the elderly. These pathologies affect the motor system in Parkinson’s or Huntington’s diseases, memory in Alzheimer’s disease, or other additional critically important neural functions after impairments in cerebral blood circulation. The retina, which is commonly considered as an outside part of the central nervous system, is also affected by neurodegenerative disorders. Most of them may lead to loss of neurons in the retinal circuitry and consecutive impaired vision or blindness. Among the neurodegenerative diseases, we mention retinal hypoxia/ischemia, glaucomatous and diabetic neuropathies, and age-related macular degeneration and those that appear less frequently, such as human recessive retinitis pigmentosa or inherited photoreceptor degeneration. These disorders may occur as a sudden pathological event or show a progressively declining clinical course, but vision may be seriously hampered or lost in all disorders. Despite different clinical symptoms, there are a number of common factors in their pathogenesis, like neuroinflammation associated with neurodegeneration. Retinal hypoxia, as a persisting insult, may induce enhanced purinergic modulation and glutamatergic–glycinergic neurotransmission. Activation of the two systems may evoke sustained release of inflammatory mediators from activated microglia and the resultant chronic proinflammatory environment may induce development or exacerbation of retinal neurodegenerative disorders. There are several excellent review articles published on different aspects of the neurodegenerative pathologies of the retina [1,2,3].

In spite of the extended research effort, the currently used therapeutics only slow down the progression of retinal neurodegenerative disorders, whereas normalization of impaired visual functions can be rarely achieved. It is, therefore, mandatory to develop novel therapeutic interventions. This goal, however, cannot be reached without unfolding the pathophysiology of retinal neurodegenerative diseases. The aim of this review is to highlight a possible interaction between purinergic and glycinergic signaling systems in neuroinflammatory–neurodegenerative disorders of the retina. This interaction is, however, further complicated by the findings that microglia, the resident immune cells of the central nervous system and retina, are involved not only in neuronal cell destructions but also in compensatory neural repair following insulting influences [4,5].

## 2. Neural Circuitries and Glial Cell Types in the Retina

Figure 1A shows the cell types and the organization of neural circuitry after hematoxylin-eosin staining of the retina [6]. The vertical section of the retina showed the retinal pigment epithelium (PRE) as the outermost layer followed by the photoreceptor layer (PRL) and outer nuclear layer (ONL), the latter containing the cell bodies of the cone and rod photoreceptors. The neurotransmitter in photoreceptors is glutamate, the release of which decreases to light exposition of the retina. The second-degree neurons in the retina are glutamatergic bipolar cells. The cell bodies of the ON and OFF cone bipolar cells and the connecting ganglion cells form the inner nuclear layer (INL) and the ganglion cell layer (GCL) of the retina, respectively. The inner nuclear layer also contains amacrine cells mostly releasing glycine or GABA as neurotransmitters, which, together with the horizontal cells, compose the horizontal visual pathway [7]. Neuronal connection, which transmits signals from rod bipolar cells to cone bipolar cells, is established by the AII cell, a specific type of glycinergic amacrine cell [8]. The outer and inner plexiform layers (OPL, IPL) contain the synaptic connections between the photoreceptors and bipolar cells and the bipolar cells and retinal ganglion cells, respectively (Figure 1B). These layers show characteristic alterations or tapering in various neurodegenerative disorders or injuries of the retina [9,10].

Besides the neural circuits, the retina also contains numerous glia cells termed macroglia (Müller cells and astrocytes) and microglial cells. The principal macroglia Müller cells span all retinal layers, whereas astrocytes are mostly located close to the nerve fiber layer and the ganglion cell layer [3,7,12]. Müller cells are the major source of adenosine triphosphate (ATP) released into the extracellular space, which then modulates both neural and microglial cell activities [7,13]. Müller cells express glycine transporter 1 (GlyT-1), which regulates extracellular glycine concentrations in the retina following its release from macroglia and amacrine cells [14,15]. The glial marker, glial fibrillary acidic protein (GFAP), is expressed in both Müller glia cells and astrocytes [13,16,17].

The other major type of retinal glia cell is microglia. They originate from macrophage or monocyte precursors and are considered as the major immunocompetent cell type of the retina [18]. In healthy retina, microglia cells are characterized by small somata and extensively ramified processes at rest [3]. Microglial cells, which exhibit different morphologies in the retina, are mainly present in the inner and outer plexiform layers, ganglion cell layer, and nerve fiber layer (Figure 1C) [7,11,19,20].

## 3. Cellular Biology of Microglia in Neural and Retinal Tissues

Microglia are the resident immune cells of the central nervous system as well as the retina. Microglia exist in different forms. Resting microglia are derived from macrophage-like cells and they can be characterized by small cell bodies and multiple branched processes arising from the cell bodies with down-regulated macrophage-like properties [21]. Microglia processes exhibit continuous movements controlling the immunological environment (surveillance mode) in order to detect signals from stressed neurons or macroglia. Microglial cells also have a function in normal retinal growth, neurogenesis, and retinal blood vessel formation under physiological conditions.

The shape of microglial cells turns to be amoeboid during microglial activation in retinopathological events and cellular hypertrophy and process retraction can be also observed [22]. The amoeboid form of microglia is enlarged in size, and microglia occur either as single cells or in clusters and their phagocytotic activity is enhanced [3]. In response to pathogenic signals such as ischemia or oxidative stress, microglia proliferate and migrate towards the site of tissue injury [23]. Pathological stimuli also activate macroglial cells and cause gliosis of the retina. The gliosis of Müller cells is characterized by cellular hypertrophy, proliferation, and upregulation of GFAP expression [13].

Ramified microglia is a model of resting-like cells, whereas addition of the bacterial cell wall component lipopolysaccharide (LPS) induces activation of microglial cells. This procedure is also called a shift of microglia from an ineffective to an effective (activated) state. Stimulation with LPS is accompanied by altered K^+^ channel expression: Resting microglial cells do not express voltage-activated K^+^ channels, their expression together with P2Y purinoceptors can be observed following microglia activation [24].

Microglial cells undergo two kinds of activation in response to infection, tissue traumatic or hypoxic injury [10,25]. These two types are designated as M1 and M2 microglial cells, which are also mentioned in the literature as proinflammatory and anti-inflammatory phenotypes [26]. The M1 first state is a neurotoxic phenotype that is characterized with massive inflammatory responses, release of tumor necrosis factor α (TNFα) and other neurotoxic inflammatory mediators/cytokines such as interleukin (IL)-1α, IL-1β, IL-6, and IL-12 [3]. Release of TNFα and IL-6 from microglial cells is toxic and induces neuronal cell death [27]. In certain circumstances, the M1 state may turn to uncontrolled activation of microglial cells, leading to a chronic inflammation in neural tissues. The increased production of the various proinflammatory cytokines may maintain an inflammatory condition and sustained release of these factors also contributes to neuronal cell death [28].

The second microglia phenotype is M2, which can secrete a series of anti-inflammatory mediators (IL-4, IL-10, IL-13, TGF-1β), and neurotrophic factors such as insulin-like growth factor 1 (IGF-1). M2 microglia serve inflammation resolution and promote neuroprotection and neuronal survival [29]. M2 microglia were recognized as neuroprotective in neurodegenerative diseases preventing neuroinflammation both in the brain and in the retina. M2 microglia downregulate inflammatory cells and the released protective and trophic factors induce immunosuppressive responses [30]. During M1 to M2 transition, the acute and prolonged phases of microglia activation may represent a dual role of these cells in neuroinflammatory and neurodegenerative disorders of the retina [26]. Chronic neuroinflammation in neural or retinal tissues may be due to lack of sufficient M2 microglia responses either because of a lower number of microglial cells or a lower secretion of neuroprotective factors [30]. It has been suggested that phenotypic shift of microglia from proinflammatory M1 function to anti-inflammatory M2 function may be beneficial in retinal neurodegenerative disorders [25].

Microglia also become activated in pathological angiogenesis in the retina. Vascular endothelial growth factor (VEGF) released from microglia and monocytes has a crucial role in blood vessel growth. VEGF is also an inflammatory mediator and contributes to inflammatory responses [31]. In a mouse model of macular degeneration, VEGF is released from microglial cells into the subretinal space causing choroidal neovascularization [7]. Increased levels of VEGF and choroidal neovascularization were reported in patients suffering from age-related macular degeneration. Elevated vitreous VEGF levels were also shown in patients with retinal vein occlusion [32]. Microglial cells can produce and the release brain-derived neurotrophic factor (BDNF) and other neurotrophic factors such as nerve growth factor (NGF), inducing trophic cascades but also cell death in developing retina [18,33,34].

## 4. Purinergic Modulation of Neural Tissues

Ischemia and other pathological conditions like mechanical perturbation and distension, cell swelling, stretching, or increased hydrostatic pressure all lead to ATP release [35]. ATP can be released from neurons into the extracellular space by an external Ca^2+^-dependent exocytotic mechanism evoked by membrane depolarization. Besides neurons, ATP can also be released from glia cells by Ca^2+^-dependent and independent means [4]. The release of ATP may also occur though the large transmembrane pannexins hemichannels from neurons and astrocytes [28,36].

ATP, if once released, is metabolized into ADP, AMP, and adenosine in the extracellular space by ecto-5′-nucleotidases (CD73) and ecto-nucleotide triphosphate diphosphohydrolases (NTPDases, CD39) [2,37,38]. Activity of these enzymes can be modified pharmacologically, e.g., ARL67156 is an inhibitor of NTPDases (Table 1). The turnover of the neural or glial ATP released towards the extracellular space may end up by uptake of the formed adenosine into neurons or glial cells. Adenosine reuptake inhibitors increase adenosine concentrations in the extracellular space (Table 1). Adenosine can also be produced intracellularly and effluxed from the intracellular space by adenosine transporter reversal. The mechanism commonly implicated in adenosine release involves bi-directional nucleoside transport. Thus, the Na^+^-dependent concentrative nucleoside transporters assure uptake of adenosine from the extracellular space (normal-mode operation), whereas operation of the transporter in the reverse or release-mode operation results in increased adenosine efflux from the cells [39]. The extracellular concentration of adenosine is also determined by the operation of the Na^+^-independent equilibrative nucleoside transporter [40]. Adenosine levels increase at the site of tissue damage or in the retina exposed to hypoxia and ischemia-reperfusion [41,42].

Purinergic influence in the retina is mediated by the release of various endogenous ligands for P1 and P2 purinoceptors from neuronal and non-neuronal cells [44,45]. In the retina, ATP release was found from the pigment epithelium layer, ganglion cells, and also from Müller glia cells and astrocytes [13,46,47]. The presence of pannexin hemichannels and their contribution to ATP release have been demonstrated in the retina [35].

### 4.1. Heterogeneity of Purinoceptors

Purinergic receptors or purinoceptors are divided into nucleoside P1 and nucleotide P2 types based upon their natural ligands. P2 purinoceptors are further divided into two groups: P2X purinoceptors are ATP-gated ion channels, whereas P2Y purinoceptors are metabotropic and coupled to G proteins. P2 purinoceptors are sensitive to adenine and guanine nucleotides [2]. Released ATP activates P2X and several P2Y purinoceptor subtypes but other adenine or uridine nucleotides ADP, UTP, and UDP are also natural agonist ligands for some P2Y purinoceptor subtypes. P2Y purinoceptors consist of eight different subtypes designated as P2Y_1,2,4,6,11,12,13,14_. The downstream signal transduction linked to P2Y purinoceptors are mostly of two kinds, either Gq (P2Y_1,2,4,6,,11_ receptors) or Gi (P2Y_12,13,14_) proteins are coupled to the receptors. Thus, P2Y purinoceptors with different signal transductions can evoke both excitation and inhibition. In addition, P2Y purinoceptors exhibit different agonist binding profiles: P2Y_1_, P2Y_12_, and P2Y_13_ purinoceptors are sensitive to adenine nucleotides; P2Y_2_ and P2Y_4_ are activated by adenine and uridine nucleotides; and human P2Y_4_ and P2Y_6_ purinoceptors are primarily affected by uridine nucleotides [23]. 

The response of Gq protein-coupled P2Y purinoceptors to ATP is an increase in intracellular inositol trisphosphate (IP_3_) level, a rapid transient release of Ca^2+^ from internal stores followed by an increase of intracellular Ca^2+^ concentration, and also a Ca^2+^ influx into the cells [34]. Illes and coworkers [48] proposed that this increase of intracellular Ca^2+^ concentrations results in opening K^+^ channels, allowing an outward-directed K^+^ current, and membrane hyperpolarization in microglial cells. 

The P2X ionotropic purinoceptors are ligand-gated cationic channels and are non-selectively permeable to Na^+^, K^+^, and Ca^2+^. P2X purinoceptors have seven subtypes identified as P2X_1_ to P2X_7_ and are sensitive to ATP but not to ADP, AMP, and adenosine [4,23]. In response to ATP, P2X purinoceptors mediate fast excitatory neurotransmission as their activation induces an inward-directed cation current and cellular membrane depolarization [34,48]. P2X purinoceptors are also linked to Ca^2+^ signaling via receptor-coupled Ca^2+^-permeable cationic channels, allowing Ca^2+^ entry into the cells [24]. 

The other types of purinoceptors are the P1 type [46]. Adenosine is the natural ligand for P1 nucleoside receptors, which activates all four different receptor subtypes, A_1_, A_2A_, A_2B_, and A_3_. Adenosine receptors are only sensitive to the nucleoside adenosine [2] whereas Luongo and coworkers [26] reported that treatment with ATP upregulated adenosine A_1_ receptor expression in microglia. Adenosine receptors possess metabotropic signal transduction: A_1_ and A_3_ receptors are coupled to Gi proteins and A_2A_ and A_2B_ receptors are linked to Gs proteins. Accordingly, activation of adenosine receptors leads to stimulation or inhibition of adenylate cyclase and to an increase or decrease of tissue cAMP levels. Activation of A_1_ receptors results in mostly opposite effects to A_2A_ receptors. The P1 purinoceptor agonist adenosine fails to generate a [Ca^2+^]i response in cultured microglia indicating that ATP action on intracellular Ca^2+^ is mediated only through activation of P2 purinoceptors [24]. Inhibitory response of purinoceptors coupled to Gi proteins (A_1_ and A_3_ adenosine receptors and some P2Y purinoceptors) is a decrease in cAMP levels leading to Ca^2+^ and K^+^ channel dephosphorylation that results in membrane hyperpolarization [26].

### 4.2. Purinoceptors in the Retina

P2 purinoceptors possess distinct expression profiles in the retina. P2Y metabotropic receptors are expressed in neurons and glial cells and the retinal pigment epithelium [49]. P2Y_1_ purinoceptors may have a preferential expression in retinal macroglia including Müller glia cells, whereas P2Y_1_, P2Y_12_, and P2Y_13_ purinoceptors were identified in astroglial cells [50]. Increased purinergic activity induces gliosis in Müller glia cells as was demonstrated by increased GFAP immunoreactivity and the involvement of P2Y_1_ purinoceptor was suggested to evoke this effect [51]. Microglial cells in the retina express the P2Y_1_ purinoceptor subtype [51].

P2X purinoceptors are expressed on most classes of neurons in the retina. The presence of P2X receptor immunoreactivity in GABAergic amacrine cells supports previous findings that purinergic signals modulate information processing by GABAergic amacrine cells [49]. Other laboratories identified an abundant expression of P2X_7_ purinoceptors in the inner and outer plexiform layers, in ganglion cells, Müller glia cells, astrocytes, and microglia in high concentrations [23,52]. Wurm and coworkers [53] reported that, although P2X_7_ receptors are expressed in human Müller cells, P2X purinoceptors are largely absent in other mammalian Müller glia cells. Depending on the receptor-coupled signal transduction, the effects mediated by P2X and P2Y receptors may be synergistic or antagonistic in retinal circuitry.

Similarly to P2 purinoceptors, P1 purinoceptors are present in retinal pigment epithelial cells and ganglion cells also express multiple adenosine receptors [49]. A_1_, A_2A_, and A_3_ adenosine receptor mRNAs were detected in the inner nuclear layer of the retina. A_2A_ and A_2B_ receptors are present in retinal pigment epithelial cells, Müller glia cells, and they are also expressed in microglia in the retina [2]. 

A1 receptors are inhibitory in nature and suppress excitatory neurotransmission [46]. Adenosine is a major inhibitor of neural activity in the retina. Increased production of adenosine in the extracellular space activates inhibitory A1 receptors and neuronal activity is decreased [13]. The A_1_ receptor agonist CPA decreased the number of apoptotic nuclei and GFAP immunoreactivity in the retina exposed to excessive light to induce neurodegeneration. This effect was reversed by the A_1_ receptor antagonist DPCPX [54]. 

### 4.3. Purinergic Regulation of Microglia

Microglia express P2Y and P2X purinoceptors in resting and activated states [4]. Further analysis demonstrated that the majority of microglial cells in proliferating (resting) state express P2X purinoceptors and non-proliferating (activated or alerted phenotype) microglial cells express both P2Y and P2X purinoceptors [48]. Färber and Kettenmann [34] reported the presence of all the eight different types of P2Y purinoceptors and six types of P2X purinoceptors (P2X_1,2,3,4,6,7,_) in the microglial cell line. Accordingly, addition of ATP affects several metabotropic and ionotropic transduction systems in microglia [55,56].

ATP and adenosine alter the morphology of microglia by inducing an outgrowth of microglia processes. ATP also regulates the motility of microglial processes and elicits rapid chemotactic responses in the retina [7,57]. Microglial chemotaxis as a response to neuronal injury is regulated by P2Y_12_ receptors. Ogata and coworkers [56] reported that activation of P2Y receptors induces microglia proliferation and accelerates process retraction of the cells. Furthermore, P2Y_1,2,4_ receptors expressed on microglia regulate phagocytosis [58]. Uckermann and coworkers [21] also reported that rabbit retinal microglial cells contain P2Y_1_ receptors and their activation leads to microglial cell activation. In addition, P2X_4_ purinoceptors mediate chemotaxis, and P2X_7_ purinoceptors are involved in transformation of microglia to the proinflammatory phenotype [26]. Of the P1 purinoceptors, A_1_ and A_3_ adenosine receptors have been negatively implicated into the regulation of microglia morphology and responses, and proliferation and mediator release [11]. A_2A_ receptor upregulation has a role in the regulation of extension and retraction of microglial processes [59].

LPS stimulated TNFα, IL-1β, and IL-6 release and ATP inhibited these effects in cultured microglia obtained from rat spinal cord. In addition, the P2Y purinoceptor agonist 2-MeSATP exerted similar inhibition [56]. It was shown that the P2Y_2_ and P2Y_6_ purinoceptor ligands UTP, UDP, and the hydrolysis-resistant ATP analogue ATPγS stimulated the basal and TNFα-induced secretion of the chemotactic factor IL-8 in human retinal pigment epithelium [60]. These observations raise the possibility that P2Y purinoceptors coupled to Gq or Gi proteins may oppositely regulate the production of proinflammatory cytokines from microglia. 

Activation of different P2 receptors occurs at a temporal scale. First, ATP activates microglial P2X_7_ purinoceptors following an ischemic insult inducing an immediate tissue necrosis followed by activation of purinoceptors on astroglia cells and neurons [5]. In addition, ATP stimulates P2X_7_ purinoceptors at milliomolar concentration, whereas P2Y purinoceptors can be activated in a 10–100 μmol concentration range of ATP [56]. P2X_7_ purinoceptors exhibit a low sensitivity to ATP resulting in activation only in pathological conditions (tissue damage, neuroinflammation, mechanical stress or injury, trauma, hypoxia, or ischemia) when extracellular ATP concentrations reach high levels. 

ATP-activated P2X purinoceptors are also principal regulators of neuroinflammatory responses. P2X purinoceptor activation induces IL-1β release from macrophages and ADP and AMP also act as promoters of LPS-induced IL-1β secretion in microglia cells [36]. P2X_7_ receptor mediates cytokine release from microglia [52]. A series of studies demonstrated that P2X_7_ receptor stimulation of immune cells by ATP evokes release of the proinflammatory TNFα and IL-1β from mouse or rat microglial cell lines [61]. This release of TNFα was found to be dependent upon Ca^2+^ influx and MAP kinase cascade activation [62]. In contrast, Shigemito-Mogami and coworkers [63] reported that the production of the proinflammatory cytokine IL-6 was under the control of P2Y rather than P2X purinoceptors in mouse microglial cell lines. 

The production and release of proinflammatory cytokines seem to be oppositely regulated by the metabotropic P2Y and the ionotropic P2X receptors in microglial cells [4]. The apparently opposite effects of P2X and P2Y purinoceptors on proinflammatory factors may be explained by the different signal transduction systems utilized by these receptors. Ogata and coworkers [56] also found opposite effects of P2X (stimulation) and P2Y purinoceptor (inhibition) on TNFα release from cultured spinal cord microglia.

Microglial cells express all subtypes of P1 purinoceptors, A_1_, A_2A//B_, and A_3_ [11]. The various types of adenosine receptors also exhibit different sensitivity to their natural ligand adenosine [39]. Gi protein-coupled A_1_ adenosine receptor stimulation decreases and Gs protein-coupled A_2A_ receptors stimulation exerts an opposite effect on microglia activity (Table 2). The A_1_ receptor agonist CPA inhibited TNFα and GFAP mRNA expression and the A_1_ receptor antagonist DPCPX caused an opposite effect on GFAP immunoreactivity in the retina [54]. CPA also lowered TNFα mRNA expression in light-induced retinal degeneration in rats. Activation of A_1_ adenosine receptors inhibited LPS-induced TNFα release in activated retinal microglial cells [2]. Madeira and coworkers [64] reported that blockade of A_2A_ receptor prevents retinal microglia activation and inflammatory responses in the retina. On the contrary, the A_2A_ receptor agonist CGS21680 decreased TNFα release in diabetic retina. CGS21680 also attenuated the expression of inflammatory IL-6 and TNFα. Suppression of the inflammatory cascade C-Raf/ERK by A_2A_ receptor activation may be involved in the latter effect in microglia [2]. Ahmad and coworkers [65] also reported that activation of A_2A_ adenosine receptors attenuated the hypoxia- or LPS-induced TNFα release. At present, conflicting reports can be found in the literature indicating that both stimulation and blockade of A_2A_ receptors regulate the release of microglial proinflammatory bioactive proteins in the same direction [11,39,66]. The A_3_ receptor agonist CF101 exerts anti-inflammatory effects and inhibits proinflammatory cytokine release (TNFα, IL-6, IL-12). Downregulation of the NF-kB signaling pathway was suggested to be involved in the mechanism of CF101 [2].

## 5. Glycinergic Transmission in the Retina

Glycine is an inhibitory neurotransmitter that exerts inhibition on various neurons by acting on strychnine-sensitive glycine receptors (GlyRs) in the retina. All the α1-4 subunits of GlyRs were identified in the mammalian retina [69]. Glycine also has a mandatory coagonist role in activation of glutamate N-methyl-D-aspartate (NMDA) receptors both in the central nervous system and in the retina [70,71,72]. 

The neural circuit of the retina contains two kinds of glycinergic interneurons. Amacrine cells in the inner nuclear layer are synapsed with the cone pathway and establish feedback inhibition of bipolar cells thereby inhibiting responses of ganglion cells to light [73]. About half of the amacrine cells are glycinergic and they form in- and output innervations with bipolar cells, ganglion cells, and other amacrine cells [74]. Glycinergic amacrine cells synapse primarily with OFF bipolar cells [75] (Figure 1B). The other type of interneuron, which operates with glycine as a neurotransmitter, is the AII amacrine cells [8,76]. These interneurons transfer the light signal from rod bipolar cells to the cone pathways. Of the retinal glial cells, Müller cells and astrocytes are the potential sources of glycine. The cellular location of glycine in the retina indicates that it serves as a neurotransmitter in neurons and a gliotransmitter in glia cells and also participates in microglia activation [77].

The release of glycine from interneurons and macroglia cells occurs with vesicular exocytosis or reverse-mode operation of glycine transporter 1 (GlyT-1). Following neuronal release, glycine may reach the extrasynaptic space by spillover mechanism and affects extrasynaptic GlyR and NMDA receptor activities. In the vicinity of these receptors, glycine concentration is determined by GlyT-1 both in the synaptic and extrasynaptic space of the retina [78]. GlyT-1 is primarily localized to glial cells (Table 3), whereas GlyT-2, which is responsible for refilling presynaptic vesicles with glycine, is absent in the retina [79,80], but see Pena-Rangel et al. [81].

Of the macroglia cells, glycine is released from both Müller glia cells and astrocytes in the retina. In energy shortage conditions, declining cellular ATP levels cause failure of ion Na+-K+-ATPase ion pump operation and as a result intracellular Na+ concentration is elevated [1]. This altered ionic milieu favors shifting the operation of both glycine and glutamate transporters into the reverse mode [82]. The consequence of transporter reversal is an increased release of the amino acid neurotransmitters glycine and glutamate as was shown in isolated rat hippocampus and retina as well [15,78]. In response to hypoxia and cellular energy failure, ATP is also released from neuronal or glial sources, although the mechanism of this release differs from that of glutamate or glycine [35]. 

### 5.1. Heterogeneity of NMDA Receptors in the Retina

The ionotropic glutamate NMDA receptors, on which glycine acts as a mandatory coagonist, exhibit uneven distribution in the retina: A great number of NMDA receptors can be found in the ganglion cell layer [9,83]. NMDA receptors are expressed in retinal ganglion cells with different subunit compositions: NR1/NR2B NMDA receptors were found on ON ganglion cells in higher density, whereas the presence of NR1/NR2A NMDA receptors was demonstrated both in ON and OFF ganglion cell dendrites [84]. NR1/NR2A receptors are believed to be localized synaptically and extrasynaptic NMDA receptors are primarily composed of NR1NR2B subunits [85]. NR1/NR2A and NR1/NR2B receptors possess different coagonists, D-serine and glycine, respectively [86]. D-Serine may be released from stressed Müller glia cells and astrocytes to potentiate the agonist effect of glutamate on synaptic NMDA receptors [87]. On the other hand, extrasynaptic glycine potentiates the effect of glutamate on extrasynaptic NR1/NR2B NMDA receptors [88]. Glycine, after its release from synapses between amacrine cells and the OFF pathway neurons, diffuses into the extrasynaptic space and influences activity of extrasynaptic NR1/NR2B receptors primarily located on ON ganglion cells [75,89]. Extrasynaptic NR1/NR2B NMDA receptors, which exhibit higher glycine sensitivity, mediate ganglion cell death by activating pro-death intracellular signaling in retinal ischemia/hypoxia [90,91].

### 5.2. Glycinergic Regulation of Microglia 

Glycine bears modulatory features on microglia and is able to influence their pro- and anti-inflammatory functions. Accordingly, Carmans and coworkers [92] reported that glycine activated macrophage functions and stimulated the production of TNFα and other proinflammatory mediators. This effect of glycine was not mediated by GlyRs but rather by activation of the neutral amino acid transporter (NAAT) system. Glycine activates Na^+^-dependent NAAT leading to increases in intracellular Na^+^ concentrations, inducing membrane depolarization, and enhancement of Ca^2+^ signaling; these events are essential for the release of inflammatory modulators [93]. Increases in extracellular glycine concentrations in response to ischemia of the retina or brain tissues may be high enough to activate the NAAT system in macrophages or microglial cells.

Schilling and Eder [77] also reported GlyR- and GlyT-independent, but NAAT-dependent, glycine-mediated effects in mouse microglia. Metabolic activity of cultured microglia was increased in the presence of glycine, whereas it was suppressed in a glycine-free medium [22]. Glycine is also known to enhance LPS-induced nitric oxide and superoxide productions in microglial cells [93]. Furthermore, glycine enhanced ATP-induced intracellular Ca^2+^ transients in microglial cells and this effect was due to Na^+^-coupled NAAT operation [93]. These effects of glycine in microglia point to a depolarization-dependent Ca^2+^ signaling and an increased production of inflammatory mediators. Ca^2+^ signals may, however, be under the control of different signaling systems in microglia. As mentioned above, ATP enhances the glycine-evoked intracellular Ca^2+^ transients in microglial cells possibly via activation of P2X purinoceptors. It was shown by Raouf and coworkers [94] that ATP initiates inflammatory cascades by acting on surface P2X purinoceptors of microglial cells.

Others concluded, however, that glycine has an inhibitory effect on immune cells by reducing the production of proinflammatory cytokines. It has been reported that glycine decreased the production of TNFα and IL-1β in human monocytes and inhibited TNFα production in rat alveolar macrophages [95,96]. These effects of glycine may be mediated by GlyR-dependent mechanisms rather than the NAAT system. These findings indicate that glycine influences macrophages by both GlyR-dependent and GlyR-independent mechanisms. 

Hayashi and coworkers [97] reported that transferred microglia- and microglia-conditioned medium potentiated NMDA receptor-mediated synaptic responses in cortical neurons. Analyses with HPLC methods revealed that glycine and serine present in microglia-conditioned medium may be responsible for potentiation of NMDA-induced currents. Released glycine accumulates around glutamatergic synapses and it may build up a concentration sufficient to induce NMDA receptor activity-mediated neurotoxicity. Further in this line, microglia, after activation, have been reported to induce NMDA receptor-mediated neurotoxicity and neuronal death [22].

In energy compromised conditions, when disruption of the balances between normal- and reverse-mode operation of GlyT-1 occurs, GlyT-1 inhibitors decrease extracellular glycine levels [15,78], i.e., this class of compounds may negatively influence microglia activity.

## 6. Neurodegenerative and Neuroinflammatory Disorders in the Retina

Of the various clinical appearances of retinopathies, hypoxia/ischemia, glaucoma, diabetic neuropathy, and age-related macular degeneration affect a great number of the population. While these disorders exhibit different symptoms, there are several common evens in their pathogeneses. Retinal hypoxia as a persisting insult may induce enhanced purinergic signaling and glutamatergic–glycinergic neurotransmission. Activation of the two systems may lead to a sustained release of inflammatory mediators from activated microglia and the resulting chronic inflammatory environment may initiate or exacerbate retinal neurodegenerative disorders. Ganglion cell destruction can be detected in parallel with abnormally elevated activity of microglia in almost all neurodegenerative disorders of the retina.

### 6.1. Enhanced Purinergic Signaling in Neurodegenerative/Neuroinflammatory Disorders of the Retina

Purinergic signaling is activated in all neurodegenerative disorders of the retina. ATP is released in excess amount into the extracellular space in hypoxia, inflammation, oxidative stress, or nutrient starvation [4]. Ischemia-like conditions trigger ATP release from rat hippocampal slices and this release likely occurs from the ischemic retina as well [35,98]. Using an in vivo microdialysis technique, co-release of ATP and glutamate was demonstrated from the accumbens nucleus following traumatic injury in the rat and interaction of ATP and glutamate has been proposed [68]. In addition, the non-selective P2 purinoceptor antagonist PPADS and the selective P2X_7_ purinoceptor antagonist BBG inhibited oxygen and glucose deprivation-induced glutamate release in rat hippocampus [99]. These findings indicate that endogenous ATP, when its release is evoked in energy-compromised conditions, may exert a stimulatory influence on glutamate release. Oxygen-glucose deprivation also evokes glutamate release from isolated rat retina [15] and glutamate released in ischemic retina evokes Ca^2+^-independent ATP release and transporter-mediated adenosine release [100]. In these experiments, hypoxia was induced by elevated intraocular pressure in rat retina and elevated glutamate release activated purinergic signaling by releasing ATP from glial cells. Thus, a bidirectional facilitation may exist between purinergic signaling and glutamatergic neurotransmission in ischemic neural and retinal tissues. 

Experimental observations suggest that purinergic signaling involves P2Y purinoceptors in the mediation of both neurodegenerative and neuroprotective processes. In mice, deletion of P2Y receptors from Müller glia and ganglion cells was associated with increased survival of amacrine cells following retinal ischemia reperfusion, whereas ischemic death of photoreceptor cells appeared to be more pronounced. P2Y purinoceptor activation likely increases cytosolic Ca^2+^-signaling during ischemia leading to neurodegeneration of amacrine cells [51]. Accordingly, P2Y purinoceptors may support survival of photoreceptors but result in death of amacrine cells. In the zebrafish retina, P2Y1 purinoceptor stimulation by ADP exerts neuroprotection of inner retinal neurons [38]. These findings raise the possibility that neurons express P2Y purinoceptors coupled to Gq or Gi proteins and the excitatory or inhibitory downstream signaling results in neurodegeneration or neuroprotection in the retina [5]. 

An increase in extracellular ATP concentration also evokes P2X_7_ purinoceptor stimulation, which opens receptor-coupled ion channels and cation influx into the cells. Activation of P2X_7_ purinoceptors, which ensures excess Ca^2+^ permeability in neurons, occurs in tissue damage, neuroinflammation, or mechanical stress. Elevated intraocular pressure is also accompanied by increased levels of ATP in the extracellular space [13]. Moreover, P2X_7_ purinoceptor activation induces release of TNFα and proinflammatory cytokines from microglia and may cause neuronal degeneration and cell death [34]. Stimulation of the P2X_7_ purinoceptor has a role in ischemic neurodegeneration in the retina and can modulate retinal ganglion cell destruction [52]. 

Increased adenosine tissue concentration was observed in rat retina following ligation of the central retinal artery [41]. This increase in tissue adenosine levels may be the consequence of increased ATP hydrolysis due to imbalance between energy supply and demand [1]. Adenosine produced from released ATP may contribute to the neuroprotective effect in ischemic retina [41]. Alternatively, energy deprivation provokes enhanced intracellular Na^+^ concentrations and equilibrative nucleoside transporter type-1 operating in the reverse-mode increases adenosine efflux [4]. Elevated extracellular adenosine may activate a series of P1 adenosine receptors expressed in retinal neurons or glial cells leading to a short-term antiischemic effect [42,101]. Accordingly, the A_1_ receptor agonist R-PIA reduced retinal damage induced by the raise in intraocular pressure and improved electroretinogram after ischemia in the rat eye [102]. On the other hand, the blockade of A_1_ receptors increases neurotoxicity caused by hypoxia/ischemia or glutamate [46]. Thus, the Gi protein-coupled A_1_ receptor signal transduction can be considered as an endogenous neuroprotective system. This neuroprotective effect may be attributed to inhibition of neuronal Ca^2+^ uptake [39]. In addition, activation of the inhibitory A_1_ receptors by adenosine attenuates the release of excitatory amino acids and exerts neuroprotection during glutamatergic neurotoxicity [46].

A_2A_ receptors, which are coupled to Gs proteins, facilitate glutamate release and also facilitate NMDA receptor function at postsynaptic levels [46]. Blockade of A_2A_ receptors with selective antagonists protect ganglion cells in retinal ischemia [11,42,101]. In microglia, inhibition of A_2A_ adenosine receptors reduces the inflammatory response in retinal pigment epithelium and also reduces photoreceptor cell death [64]. In addition, the adenosine A_2A_ receptor antagonist SCH58261 prevented microglia-mediated neuroinflammation and induced neuroprotection in the retina [64]. However, a series of experiments is paradoxical: It has been reported that the A_2A_ receptor agonist CGS21680, similarly to the receptor antagonist, decreased microglial activation and retinal cell death in a mouse traumatic optic neuropathy model [65], but see Santiago et al. [66].

### 6.2. Enhanced Glutamatergic–Glycinergic Tone in Neurodegenerative/Neuroinflammatory Disorders of the Retina

The concentration of glutamate, the principal neurotransmitter in the retina, is elevated in the vitreous body or aqueosus humor following experimental retinal ischemia, optic nerve neuropathy, and also in glaucoma patients [103]. Increases in extracellular glutamate concentration facilitate glutamate uptake into Müller glia cells, leading to cellular swelling [100]. This finding points to the important role of glia cells in neurodegenerative disorders including excitotoxic injury of the retina [1,103]. In addition, NMDA receptors are overstimulated when extracellular glutamate concentration is increased and this increase plays a pivotal role in neurodegenerative disorders of the retina. The resulting cellular insults can be ameliorated by using various NMDA receptor antagonists [1,104], albeit their clinical usefulness is a subject of debate.

Increased glycine levels have also been demonstrated in neurodegenerative–neuroinflammatory disorders of the central nervous system (amyotrophic lateral sclerosis, multiple sclerosis) and also of the eye [92]. Elevated glycine concentrations were found in the vitreous body in ischemia following ophthalmic artery occlusion [105]. Exocytotic glycine release can also be demonstrated from glycinergic amacrine cells and this release may be directed outward concerning the synaptic cleft by the spillover mechanism. Glycine may be released from Müller glia cells and astrocyte by reverse-mode operation of GlyT-1 [71]. 

In physiological conditions, the GlyT-1-mediated glycine uptake reduces the possibility of glycine/glutamate-induced neurotoxicity. In oxygen and glucose deprivation-induced neurotoxicity, a decrease in glycine uptake and an increase in glycine release were found to indicate a forced reverse-mode operation of GyT-1 in rat retina [15]. Ischemic conditions evoke simultaneous increase in [^3^H]glutamate and [^3^H]glycine release in the retina ensuring excess availability of the agonist and co-agonist at NR1/NR2B-type NMDA receptors [15]. The coagonist role of glycine in NMDA receptor activation indicates that glycine concentrations determined by GyT-1 may have a key role in neurodegenerative disorders of the retina [15,71].

### 6.3. Activated Microglial Cells in Neurodegenerative Disorders of the Retina 

Elevated intraocular pressure, which results in hypoxia in the retina, elicits glaucomatous neuropathy characterized by damage of ganglion cell axons and gradual loss of ganglion cells [100,106]. Reactive microglia have been observed in the retina following increased ocular pressure [11]. Microglial cells are redistributed in the retina as they can be detected in close vicinity of ganglion cell soma and axons [10]. Activation of microglia is showed by the observed increases of TNFα, IL-6, and IL-8 proinflammatory cytokines in aquesous humor obtained from glaucoma patients [11]. These findings indicate that a neuroinflammatory component is present and plays a significant role in glaucomatous retinal neuropathy. 

Retinal hypoxia may also be involved in the pathomechanism of diabetic neuropathy of the retina [51]. Tissue hypoxia may result from insufficient microvascular circulation in diabetic neuropathy, which is accompanied by neurodegenerative events in the neural network of the retina. Reactive microglia in diabetic retinopathy are hypertrophic and exhibit an amoeboid shape. The perivascular presence of reactive microglia can be detected in cluster formation in diabetic retina [7]. Increased cytokine levels (TNF, IL-1β, IL-6, and IL-8) have been reported in the vitreous fluid of the eye indicating the involvement of a chronic inflammatory process in diabetic retinopathy [3]. Microcirculatory failure also induces a release of VEGF in diabetic retinopathy [107].

In age-related macular degeneration, pathological changes can be found in the retinal pigment epithelium and loss of photoreceptors is a key finding [64]. Impairment of photoreceptor functions is a consequence of neovascularization as novel blood vessels are formed in the outer retina in the wet-form of this retinal pathology. In age-related macular degeneration, microglia are accumulated in the outer nuclear layer and the subretinal space and around drusen deposits. Choroidal neovascularization and degeneration of the retinal pigment epithelium and photoreceptors are further characteristics of this disorder. Microglia are enlarged and exhibit an amoeboid shape in the retinal pigment epithelium. Activated microglia induce secretion of the proinflammatory cytokine IL-1β and increased cytokine levels were reported in the aqueosus humor [3]. In wet-form of age-related macular degeneration, hypoxia/ischemia induces expression of VEGF, which results in disruption of the blood–retinal barrier and retinal edema underlies the pathology of macular degeneration [31].

## 7. A Tripartite Interaction: Purinergic–Glycinergic Cross-Talk and Microglia Activation in Neurodegenerative–Neuroinflammatory Disorders of the Retina

We have hypothesized a tripartite interaction in the development of neuroinflammatory–neurodegenerative disorders of the retina with the participation of the purinergic signaling, glutamatergic–glycinergic neurotransmission, and activated microglial cells. This interaction may be synergistic or attenuative depending on whether the participating events strengthen or debilitate each other’s effects, creating positive or negative feedback regulation. Whether pathological changes reinforce or weaken the functions of the participants in the tripartite interaction may have importance in microglia activation. Thus, microglia in an M1-activated state may turn into a sustained activation and chronic inflammatory pathologies in the retina, whereas microglia turning into an M2 state induce neuroregeneration, remodeling, and neuronal repair [30].

Activation of the tripartite interaction may be triggered by anoxia/ischemia resulting from disturbances in retinal blood circulation [31,52]. Thromboembolic occlusion of the retinal artery or vein, microcirculatory abnormalities in diabetic neuropathy, increased intraocular pressure, pathological angiogenesis in age-related macular degeneration, or arteriosclerosis of arteries in the optic nerve head all lead to reduced arterial blood flow in the posterior eye [32,105,108]. These disturbances in blood circulation lead to cellular energy shortage, which is accompanied by increased release of ATP and the amino acid neurotransmitters glycine and glutamate from neuronal or glial sources. ATP, when released in response to hypoxia, stimulates P2X receptors expressed on resting microglial cells facilitating their transit to activated state and neuroinflammatory conditions develop. Microglial cell activation by purinergic signaling may result in a sustained pathological state as ATP evokes glutamate release and in turn glutamate stimulates further efflux of ATP [28]. Thus, there might be a self-strengthening regulation between ATP and glutamate release occurring in hypoxic conditions. Further in line with glutamate released from neurons or macroglia cells in high concentrations activates NMDA receptors leading to a condition when excitotoxicity may be triggered.

Hypoxic conditions, however, also evoke glycine release from neurons and astrocytes in the central nervous system and retina. High extracellular glycine concentration stimulates microglia activity and the glycine-induced proinflammatory cytokine release may be an additional factor in neuronal cell damage in retinal circuitry [34,100]. Moreover, high extracellular glycine levels ensure the required coagonist concentration for activation of NMDA receptors. Microglial glycine has been reported to participate in stimulation of NMDA receptors leading to the possibility that activated microglial cells evoke in part overactive NMDA receptor-induced neurodegeneration [22]. Thus, interaction among neurons, macroglia, and microglial cells in glycine release also forms a self-strengthening regulation, which may end up in neurodegeneration.

Besides these self-strengthening mechanisms, attenuative regulations can also be observed in the proposed tripartite interaction. Accordingly, activated microglia are altered in order to express purinoceptors of the P2Y type and their activation decreases the production and release of proinflammatory cytokines and progress of neuroinflammation may slow down [56]. Moreover, P2Y purinoceptors activated by ATP on astrocytes stimulate normal (uptake) mode operation of GlyT-1 leading to a decrease of extracellular glycine concentrations [50]. Recognition of the coagonist role of glycine in glutamate-mediated NMDA receptor activation led to the conclusion that decreases in extracellular glycine concentration may actually result in NMDA receptor hypofunctionality and exert neuroprotection in retinal neurodegenerative disorders. This goal can also be reached by inhibition of the reverse-mode operation of GlyT-1 by its selective inhibitors [15,109]. As shown in Figure 2A, the GlyT-1 inhibitor ACPPB decreased the oxygen-glucose deprivation-induced [^3^H]glycine release in isolated rat retina and this effect was due to an inhibition of the reverse-mode operation of GlyT-1. Besides inhibition of the reverse mode operation of GlyT-1, stimulation of normal mode operation also leads to a decrease of extracellular glycine concentrations and consequently to inhibition of overactivated NMDA receptors.

Jiménez [50] and coworkers reported that the P2Y_1,12,13_ purinoceptor agonist 2-MeS-ADP stimulated the uptake (normal) mode operation of GlyT-1 in rat brainstem primary neuronal cultures. This effect was partially reversed by the P2Y_1_ antagonist MRS2179, the P2Y_13_ antagonist MRS2211, and the non-selective P2 purinoceptor antagonist suramin (Table 1). This finding suggests the involvement of the Gq protein-coupled P2Y_1_ and Gi protein-coupled P2Y_13_ purinoceptors in the regulation of GlyT-1 activity with an apparent P2Y_1_ purinoceptor dominance. Thus, besides GlyT-1 inhibition, P2 purinoceptor stimulation may also decrease extracellular glycine concentrations by stimulation of the normal-mode operation of GlyT-1. As shown in Figure 2B, the non-selective P2 natural purinoceptor agonist ATP decreased the oxygen-glucose deprivation-induced [^3^H]glycine release from rat retina. While we consider our finding preliminary and in need of further confirmation using more selective receptor ligands, we speculated that ATP stimulating P2Y purinoceptors forces GlyT-1 operation into the uptake mode. The resulting decrease in extracellular glycine concentrations may then prevent overactivation of NMDA receptors and thereby attenuate the exocytotic cascade. The biological substrate for the cross-talk between purinoceptors and GlyT-1 might be the astrocytes, which express P2 type purinoceptors as well as GlyT-1 immunoreactivity in the retina [68,79].

The third participant in this tripartite interaction is microglia. Concerning retinal pathologies, in which glutamatergic–glycinergic neurotransmission and ATP/adenosine-mediated signaling are functional, the presence of activated microglia has also been demonstrated. Induction of proinflammatory cytokine production may be deleterious to injured cells after ischemic insults as was shown in retinal vascular occlusion and reperfusion [112]. Microglial cells in the hypothesized tripartite interaction receive stimulation by glycine released in excess from neurons and astrocytes and by ATP and adenosine via P2X purinoceptors and stimulatory adenosine receptors. On the other hand, a number of experiments point to the inhibitory purinergic influence mediated by ATP and adenosine via P2Y purinoceptors and inhibitory adenosine receptors. The balance between stimulatory and inhibitory influences determines whether activated microglia are shifted to permanent neuroinflammation or remodeling and neuronal repair occur. The hypothesized tripartite interaction between glutamatergic–glycinergic neurotransmission, purinergic modulation, and microglia is shown in Figure 3.

## 8. Conclusions: Possible Therapeutic Consequences

The tripartite interaction between purinergic modulation, glutamatergic–glycinergic neurotransmission, and microglia activation provides three therapeutic target pathways in neuroinflammatory–neurodegenerative disorders of the retina. First, GlyT-1 inhibitors [16,113] block reverse-mode operation of the transporter in pathological conditions and decrease the concentrations of the coagonist glycine at NR1/NR2B-type NMDA receptors. These effects of GlyT-1 inhibitors may lead to protection of retina ganglion cells. Moreover, decreased extracellular glycine concentrations may also lead to decreased microglia activity with therapeutic consequences.

Second, P2Y purinoceptor agonists and P2X purinoceptor antagonists inhibit micoglial activity and decrease the production and release of TNFα and other proinflammatory biological substances. Thus, modulation of microglial reactivity by P2Y agonists and P2X antagonists has emerged as a possible therapeutic intervention to maintain neurodegenerative courses at a low level. A similar effect might be expected from A_1_ adenosine receptor agonists and A_2_ adenosine receptor antagonists [43].

Third, microglia activation inhibited by minocycline may suppress the damage of retinal ganglion cells in ischemia and glaucoma [114]. Moreover, decreasing circulated VEGF released from microglia or blockade of VEGF receptor may slow down exacerbation of choroidal neovascularization with a beneficial influence on retina functions in diabetic retinopathy and wet-form of retina macular degeneration [7,31].

## Figures and Tables

**Figure 1 ijms-22-06209-f001:**
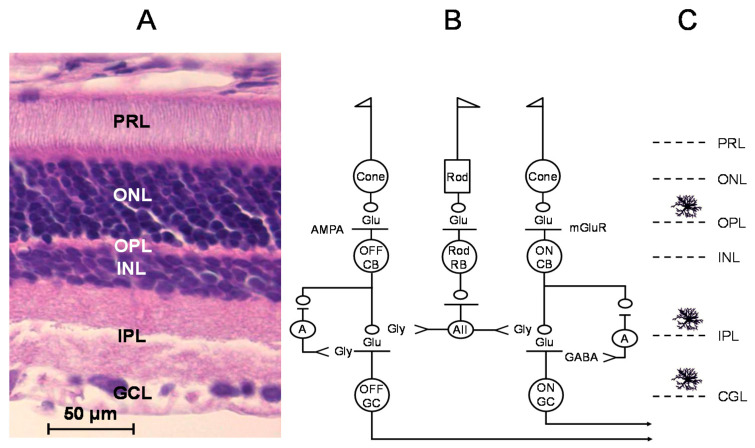
(**A**) Cytoarchitecture of the retina. Vertical section of rat retina stained with hematoxylin-eosine (made with the courtesy of Dr. Mihaly Albert). (**B**) Neural circuitry of the retina. Cone bipolar (CB) axons descend to the inner plexiform layer; the two categories of bipolar cells terminate at different levels: Axons of the OFF bipolar cells send axons to the upper part of the inner plexiform layer and the ON bipolar cells end in the lower part. Rod bipolar (RB) cells release glutamate and synapse to AII amacrine cells. Glycinergic AII cells append to OFF bipolar terminals and OFF ganglion cell dendrites in the upper level of the inner plexiform layer and these interneurons also synapse to ON cone bipolar cells in the lower half of the inner plexiform layer. Retinal ganglion cells receive excitatory glutamatergic innervation from cone bipolar cells and inhibitory influence from amacrine cells. About half of the amacrine cells are glycinergic and the other half is GABAergic. The OFF pathway is under the direct control of glycine released from glycinergic amacrine cells, whereas the ON pathway is under the inhibition of GABAergic amacrine cells. NR1/NR2A-type NMDA receptors are present in the synapses formed between bipolar and ganglion cells, whereas extrasynaptic NR1/NR2B-type NMDA receptors are predominantly expressed in ON ganglion cells. Thus, the primary target for excitotoxicity is the ON ganglion cells in the retina. (**C**) Microglial cells in normal retina are mainly distributed in the outer and inner plexiform layers. In the glaucomatous retina, microglia are present in the ganglion cell layer surrounding the retinal ganglion cell axons and soma. In the diabetic retina, perivascular accumulation of activated microglia cells can be found. These cells are accumulated in the outer nuclear layer and subretinal space in age-related macular degeneration [7,10,11]. PRL, photoreceptor layer; ONL, outer nuclear layer; OPL, outer plexiform layer; INL, inner nuclear layer; IPL, inner plexiform layer; GCL, ganglion cell layer; A, amacrine cell; AII, AII glycinergic interneuron; AMPA, AMPA receptor; mGluR, metabotropic glutamate receptor; CB, cone bipolar cell; RB, rod bipolar cell; GC, ganglion cell.

**Figure 2 ijms-22-06209-f002:**
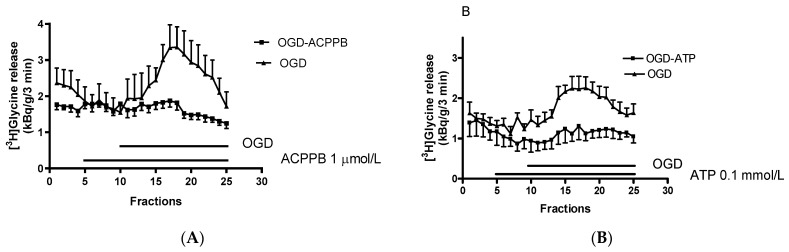
Oxygen and glucose deprivation (OGD)-induced [^3^H]glycine release from rat retina. This release was reversed by addition of the GlyT-1 inhibitor ACPPB (**A**) and the P2 purinoceptor agonist ligand ATP (**B**). Posterior eyecups containing the retinae were prepared from male Wistar rats, loaded with 10 µCi [^3^H]glycine for 30 min, and perfused with Krebs-bicarbonate buffer aerated with 95% O_2_/5% CO_2_ gas mixture for 60 min; then 25 three-min fractions were collected. The perfusion rate was kept at 1 mL/min; [^3^H]Glycine in the collected fractions and the tissue at the end of superfusion was determined and expressed as kBq/g/3 min. To evoke [^3^H]glycine release, the eyecups were superfused with glucose-free Krebs-bicarbonate buffer saturated with 95% N_2_/5% CO_2_ gas mixture added from fraction 10 and maintained through the experiment. The GlyT-1 inhibitor ACPPB, (Merck 13-h, glycine uptake inhibition IC_50_ 1.1 × 10^−8^ mol/L was determined in rat cerebral cortex synaptosomes) was added in a concentration of 10^−6^ mol/L from fraction 5 and maintained through the experiment [110,111]. ACPPB was synthesized by Professor Dr. Peter Matyus, Semmelweis University, Budapest, Hungary. ATP was added in a concentration of 10^−4^ mol/L from fraction 5 and maintained through the experiment. Data shown as mean±S.E.M., *n* = 3–4. For methodological details see Hanuska et al. [15].

**Figure 3 ijms-22-06209-f003:**
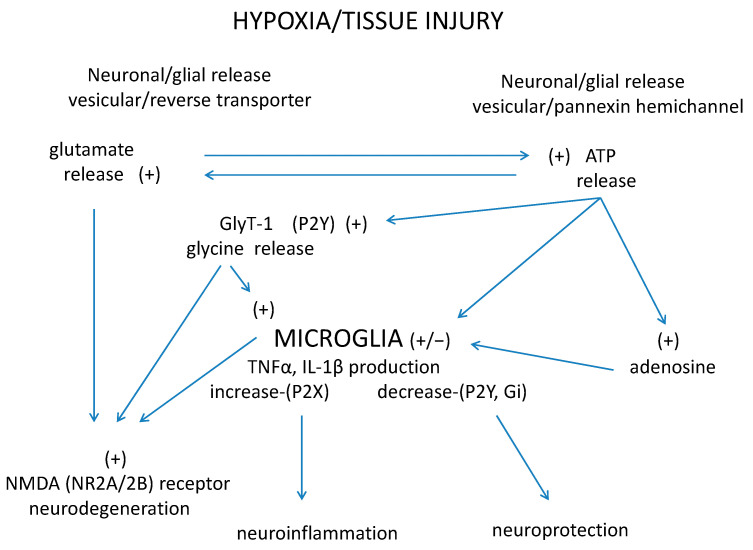
Purinergic–glycinergic cross-talk and microglia activation in neurodegenerative–neuroinflammatory disorders in the retina: A hypothetical model. Impaired microcirculation evokes energy deficiency in the retina leading to increased release of glutamate from neurons or macroglial cells by vesicular exocytosis or reverse-mode operation of excitatory amino acid transporter (EAAT). Cellular energy deficiency also evokes increase in ATP efflux from neurons or glia cells by Ca^2+^-dependent exocytosis or opening of pannexin hemichannels. There may be a self-strengthening interaction between glutamate and ATP: Glutamate induces ATP release and ATP efflux leads to an increase of glutamate release. In addition, glycine is released from energy-compromised neurons and macroglia by reverse-mode operation of glycine transporter 1 (GlyT-1) in the retina. Excess release of glycine and glutamate overactivates extrasynaptic NR1/NR2B type NMDA receptors, which evokes neurotoxicity. In contrast, synaptic NR1/NR2A type NMDA receptors mediate neuroprotection. Increased glycine release induces further release of glycine from activated microglial cells by altering Ca^2+^ transients and shifts neutral amino acid transporter (NAAT) operation in reverse-mode. Glycine-induced glycine release from microglia, a self-strengthening interaction, facilitates glutamate-induced overstimulation of NMDA receptors, triggering excytotoxicity. This mechanism can be an additional self-strengthening interaction: Glycine induces further release of glycine from activated microglial cells, which participates in extrasynaptic NMDA receptor overstimulation. The evoked NMDA receptor-mediated neurotoxicity primarily damages ON ganglion cells in the retina. Further in the interactions: ATP released by cellular energy deficiency stimulates GlyT-1 normal operation via activation of P2Y purinoceptors, which decreases extracellular glycine levels. GlyT-1 inhibitors inhibit pathological reverse-mode operation of GlyT-1. These two effects on GlyT-1 operation inhibit microglia activity by decreasing extracellular glycine concentrations. ATP released from stressed neurons and glia cells stimulates P2X and P2Y purinoceptors expressed in microglia, which induces stimulation or inhibition of the production and release of TNFα and proinflammatory cytokines (IL-1α, IL-1β, IL-6, IL-12). Adenosine produced by a breakdown of ATP or released from cells stimulates or inhibits microglia activity via adenosine receptors; the participating receptors with opposite effects are the inhibitory A_1_ and A_3_ and the stimulatory A_2A_ and A_2B_ adenosine receptors. The balance of this tripartite interaction may either be shifted to a neurodegenerative–neuroinflammatory direction or lead to neuroprotection serving inflammation resolution and neuronal survival.

**Table 1 ijms-22-06209-t001:** Drugs used to investigate purinoceptors in neurodegenerative–neuroinflammatory disorders of the central nervous system and retina.

Drugs	Mode of Action
P2 nucleotide purinoceptors	
α,β-Methylene-ATP	P2 agonist
PPADS	P2 antagonist
Suramin	non-specific P2 antagonist
P2Y nucleotide purinoceptors	
2-MeS-ATP	P2 agonist
2-MeS-ADP	P2Y_1,12,13_ agonist
MRS 2365	P2Y_1_ agonist
MRS 2179	P2Y_1_ antagonist
MRS 2211	P2Y_13_ antagonist
P2X nucleotide purinoceptors	
β,γ-Methylene-ATP	P2X agonist
BzATP	P2X_7_ agonist
TNP-ATP	P2X antagonist
NF449	P2X_1_ antagonist
Brillant Blue G (BBG)	P2X_7_ antagonist
P1 nucleoside (adenosine) purinoceptors	
R-PIA	A_1_ agonist
CCPA	A_1_ agonist
CPA	A_1_ agonist
DPCPX	A_1_ antagonist
CGS21680	A_2A_ agonist
SCH58261	A_2A_ antagonist
ZM-241,385	A_2A_ antagonist
BAY 60-6583	A_2B_ agonist
LUF-5835	A_2B_ agonist
MRS-1706	A_2B_ antagonist
CF101	A_3_ agonist
CP-532,903	A_3_ agonist
MRE 3008F20	A_3_ antagonist
Other drugs influencing purinergic signaling	
ARL67156	ecto-ATPase inhibitor
NBMPR	adenosine reuptake inhibitor
Abbreviations	
ARL 67156	6-N,N-diethyl-D-β,γ-dibromomethylene ATP
BAY 60-6583	2-[[6-Amino-3,5-dicyano-4-[4-(cyclopropylmethoxy)phenyl]-2-pyridinyl]thio] acetamide
BzATP	2′,5′-O-4-benzo-yl)-ATP
CCPA	2-chloro-N6-cyclopentyladenosine
CF101	(N6-(3-iodobenzyl)-5′-N-methylcarboxamidoadenosine
CGS 21680	4-[2][6-amino-9-(N-ethyl-β-D-ribofuranuronamidosyl)-9H-purin-2-yl]amino]ethyl] benzeneproprionic acid
CP-532,903	(2S,3S,4R,5R)-3-amino-5-[6-(2,5-dichlorobenzylamino)purin-9-yl]-4 hydroxytetrahydrofuran-2-carboxylic acid methylamide
CPA	N6-cyclopentyladenosine
DPCPX	Dipropylcyclopenthylxanthine
LUF-5835	1-[6-amino-9-[(2R,3R,4S,5R)-3,4-dihydroxy-5-(hydroxymethyl)oxolan-2-yl]purin2-yl]-N-methylpyrazole-4-carboxamide
2-MeS-ATP	2-methylthio-ATP
2-MeSADP	2-methylthio-ADP
MRE 3008F20	N-[2-(2-Furanyl)-8-propyl-8H-pyrazolo[4,3-e][1,2,4]triazolo[1,5-c]pyrimidin 5-yl]-N′-(4-methoxyphenyl)urea
MRS 1706	N-(4-acetylphenyl)-2-([4-(2,3,6,7-tetrahydro-2,6-dioxo-1,3-dipropyl-1H-purin-8-yl)phenoxy]acetamide
MRS 2179	N6-methyl-2′-deoxyadenosine-3′,5′-bisphosphate
MRS 2211	pyridoxal-5′-phosphate-6-azo(2-chloro-5-nitrophenyl)-2,4-disulfonate
MRS 2365	[[(1R,2R,3S,4R,5S)-4-[6-amino-2-(methylthio)-9H-purin-9-yl]-2,3-dihydroxy bicyclo-[3.1.0]hex-1-yl]methyl] diphosphoric acid monoester
NBMPR	S6-(4-nitrobenzyl)mercaptopurine riboside
NF449	4,4′,4′′,4′′′-[Carbonylbis(imino-5,1,3,-benzenetriyl-bis(carbonylimino))]tetrakis-1,3 benzenedisulfonic acid
PPADS	pyridoxalphosphate-6-azaphenyl-2,4-disulfonic acid
R-PIA	R-N6-(2-phenylisopropyl)adenosine
SCH5826	1,7-(2-phenylethyl)-5-amino-2-(2-furyl)-pyrazolo[4,3-e]-1,2,4-triazolo[1,5-c]pyrimidine
TNP-ATP	2′,3′-(2,4,6,-trinitrophenyl)adenosine-5′-triphosphate
ZM241,385	4-(2-[7-amino-2-)2-furyl(triazolo-[1,3,5]triazin-5-ylamino]ethyl)phenol

For further details see Jacobson and Civan [43].

**Table 2 ijms-22-06209-t002:** Purinoceptors, their signal transductions and endogenous ligands.

Receptor	Signal Transduction	Ligands
P2Y nucleotide purinoceptors		
P2Y purinoceptors subtypes		
P2Y_1,2,4,6,11*_	G_q_ protein-coupled	ATP, ADP and/or UTP, UDP
P2Y purinoceptors subtypes		
P2Y_12,13,14_	G_i_ protein-coupled	ADP, UDP
P2X nucleotide purinoceptors		
P2X purinoceptors subtypes		
P2X_1,2,3,4,5,6,7_	cationic ion channel-coupled	ATP
P1 nucleoside purinoceptors		
A_1_ adenosine receptor	G_i_ protein-coupled	adenosine
A_2A_, A_2B_ adenosine receptor	G_s_ protein-coupled	adenosine
A_3_ adenosine receptor	G_i_ protein-coupled	adenosine

P2Y_1,12,13_ purinoceptors are sensitive to adenine nucleotides, P2Y_2,4_ purinoceptors are activated by adenine and uridine nucleotides, human P2Y_24,6_ are affected primarily by uridine nucleotides [4,23], P2Y_11*_ purinoceptors also mediate stimulation of adenylyl cyclase activation. Increased cAMP may mobilize intracellular Ca^2+^ via activation of PLC-epsilon [67,68].

**Table 3 ijms-22-06209-t003:** Some characteristics of glia cells in the retina.

Glia Cell Type	Locations in the Retina	Receptor Subtype Expression	GlyT Expression
Macroglia			
Müller cells	Span all retina layers	P2Y_1_	GlyT-1
		P2X_7_	
		A_2A/2B_	
Astroglia	Ganglion cell layer	P2Y_1,12,13_	GlyT-1
		P2X_7_	
Microglia			
In healthy retina	Inner and outer plexiform layers	P2X_4,7_ A_2A/2B_	?
In retina pathologies	Ganglion cell layer	P2Y_1,2,4,12_	?
	Perivascular accumulation	P2X_7_	
	Outer nuclear layer	A_1,2A/2B,3_	
	Subretinal space		

?-no data about GlyT in microglia

## Data Availability

Not applicable.

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
