# Peer review of "Purinergic–Glycinergic Interaction in Neurodegenerative and Neuroinflammatory Disorders of the Retina"

_ijms, 2021, doi:10.3390/ijms22126209_

Round 1
Reviewer 1 Report
Dr. Laszlo G. Harsing, Jr. et al. presented a comprehensive, well-organized introduction containing metabolic mechanisms and clinic relevance in neurodegenerative and neuroinflammatory disorders of the retina. I did not have any major concerns, only several minor issues listed below:
- Please provide the scale bar for Figure 1A.
- Table 1 contains two tables, one is for introducing the drugs and another is for their full name, please label.
- Line 596, it seems that the sentence is incomplete.
- I suggest the authors could try to summarize the main feature in Part 4&5, like Figure 3 for Part 7.
Author Response
Referee 1 We appreciate the valuable comments of Referee 1.
Fig.1A. Scale bar is added to the figure.
The form of Table 1 is modified according to Referee’s suggestion.
The have checked the sentence in line 596.
An additional table (Table 3) is now added to link Parts 4 and 5.
Reviewer 2 Report
The authors have undertaken a fairly comprehensive review of neurodegenerative-neuroinflammatory disorders of the retina. The authors concluded that the tripartite interaction between purinergic modulation, glutamatergic/glycinergic neurotransmission, and microglia activation provides three therapeutic targets, such as glycine transporter 1 inhibitors, P2Y purinoceptor agonists and P2X purinoceptor antagonists, and microglia inhibition by minocycline. I have some comments that I believe need to be addressed prior to publication of this article.
Comments:
Page 8 lines 164–167, “The response of P2X purinoceptors to ATP mediates fast excitatory neurotransmission as their activation induces an inward-directed K+ current and cellular membrane depolarization [34,48].” Generally, activation of K+ currents induce membrane hyperpolarization. The P2X ionotropic purinoceptors are non-selectively permeable to cation. Thus, the activation of P2X purinoceptors induces an inward-directed cation current.
Page 10 lines 387–389, “Suppression of the inflammatory cascade C-Raf/ERK by A2A receptor activation may be involved in the latter effect in microglia (Guzman-Aranguez et al., 2014).”, Please show the reference number.
Page 11 line 402, “P2Y11* purinoceptors alsomediate stimulation of adenylyl cyclase activation.”, Please revise this sentence.
Author Response
Referee 2 We thank Referee2 for the constructive comments.
Page 8, lines 164-167. Sentence is corrected as was suggested: inward-directed cation current.
Page 10, lines 367-389. Reference Guzman-Aranguez is numbered in page 10 para 4.
Page 11, linel 402. Table 2. footnote is corrected.